# Evaluation of Nasal Microbial Communities of Beef Calves During Pre-Weaning Outbreak of Bovine Respiratory Disease

**DOI:** 10.3390/ani15192914

**Published:** 2025-10-07

**Authors:** Amy N. Abrams, Larry A. Kuehn, John W. Keele, Michael G. Gonda, Tara G. McDaneld

**Affiliations:** 1Department of Animal Science, Berry College, Mount Berry, GA 30149, USA; abrams.amyn@gmail.com; 2United States Department of Agriculture, Agricultural Research Service, U.S. Meat Animal Research Center, Clay Center, NE 68933, USA; larry.kuehn@usda.gov (L.A.K.); john.keele@usda.gov (J.W.K.); 3Department of Animal Science, South Dakota State University, Brookings, SD 57007, USA; michael.gonda@sdstate.edu

**Keywords:** cattle, 16S, microbiome, nasal, respiratory disease

## Abstract

**Simple Summary:**

Bovine respiratory disease is one of the most costly and common health problems affecting young beef calves, causing sickness, death, and significant economic losses for cattle producers. While many bacteria live naturally in the nasal cavity of healthy calves, some can become harmful under stressful conditions and trigger disease outbreaks. This study examined the nasal microbiome of calves during two respiratory disease outbreaks before weaning, one moderate and one more severe, and then again one month after treatment. We found that calves in the more severe outbreak had less variety in their nasal bacteria compared to calves in the moderate outbreak. In both groups, bacterial diversity increased after treatment, suggesting recovery of a healthier microbial balance. During outbreaks, certain harmful *Mycoplasma* species were most prominent. These findings suggest that a loss of bacterial diversity and an overgrowth of harmful *Mycoplasma* are linked to respiratory disease in calves. Understanding how these microbial communities shift during disease may help develop better ways to prevent and manage respiratory disease in cattle, ultimately improving animal health and reducing economic losses.

**Abstract:**

Bovine respiratory disease complex (BRDC) is a leading cause of morbidity and mortality in pre-weaned calves, yet the role of commensal nasal microbiota in outbreak severity remains poorly understood. This study characterized nasal bacterial communities during two BRDC outbreaks of differing severity (moderate vs. severe) and at ~30 days post-treatment. Nasal swabs were collected from calves and analyzed using 16S rRNA gene sequencing (V1–V3 regions, Illumina MiSeq) and quantitative PCR targeting three major BRDC pathogens. Microbial community profiles differed between outbreak groups and across timepoints. Calves in the severe outbreak group exhibited lower microbial diversity compared to those in the moderate outbreak. In both groups, diversity significantly increased from outbreak to post-treatment. At the time of disease, nasal communities were dominated by the genera *Mycoplasmopsis*, *Mesomycoplasma*, and *Caviibacter*, with qPCR confirming *Mycoplasma bovirhinis* as the predominant species. These findings indicate that BRDC outbreaks in pre-weaned calves are associated with reduced microbial diversity and the dominance of pathogenic *Mycoplasma* species, with recovery characterized by greater bacterial diversity. Shifts in nasal microbiome composition between outbreak and post-treatment may reflect pathogen-driven disruption during disease and subsequent microbial community rebalancing.

## 1. Introduction

Bovine respiratory disease complex (BRDC) is the most prominent disease in the North American beef cattle industry. Despite improvements in vaccination and management practices, BRDC continues to be the leading cause of morbidity and mortality in the feedlot. It has been estimated that economic loss due to BRDC is over USD 1 billion annually when considering the combined cost of prevention and treatment [1,2]. Although much of the research surrounding BRDC focuses on the feedlot phase, BRDC in pre-weaned calves accounts for losses of approximately USD 165 million annually to cow–calf producers [3]. The multifaceted nature of BRDC creates challenges in the prevention, diagnosis, and treatment of this disease. The occurrence of BRDC is dependent on complex interactions between host, pathogens, and environmental factors [4,5,6]. Included in these factors are age, breed, weight, dramatic changes in temperature, humidity, and management practices that increase stress (weaning, transportation, handling, dehorning, castration, comingling, nutritional changes, etc.) [7,8,9,10,11]. In pre-weaned calves, BRDC often presents with coughing, nasal discharge, fever, labored breathing, reduced appetite, lethargy, and death [1,2,3,5]. This results in both direct cost to producers (calf loss, vaccines, labor) and indirect cost (poor weight gain and lack of uniformity in calves) [3].

Many bacterial species associated with BRDC are considered common inhabitants of the bovine respiratory tract in healthy animals [12,13]. It is likely that diverse microbial communities in healthy animals can suppress pathogenic bacteria from colonizing in the bovine respiratory tract, but when this symbiosis is disrupted, the opportunistic pathogenic bacteria are able to proliferate [12,14,15]. Because BRDC pathogens spread readily within groups of cattle, producers commonly use mass antimicrobial treatment (metaphylaxis) for at-risk cohorts at stressful timepoints or during outbreaks to substantially reduce BRDC morbidity and mortality in high-risk cattle [16]. However, mass treatment increases overall antimicrobial use and has been associated with higher detection of multidrug-resistant BRDC bacteria, raising concerns about antibiotic overuse, added costs to producers, and contribution to the emergence and spread of antimicrobial resistance in cattle as well as indirectly to humans through the food supply and environmental transmission [17,18,19]. Thus, a better understanding of the relationship between microbial community dysbiosis and the development of BRDC could contribute to improved detection, mitigation, and treatment strategies. 

To date, research surrounding bacterial pathogens associated with BRDC has primarily focused on outbreaks associated with the feedlot period after weaning [6,12,14,15,20]. Bovine respiratory disease complex is most prevalent during the feedlot period, and animals may be predisposed to BRDC based on bacterial inhabitants of the microbiome of the upper nasal cavity. Therefore, characterizing the upper nasal microbiome during a BRDC outbreak prior to entry into the feedlot at weaning could provide insights into the role of microbial diversity, dysbiosis, and pathogenic bacterial interactions in the development of BRDC. This study aimed to characterize the nasal microbiome of calves at the U.S. Meat Animal Research Center (USMARC) during a pre-weaning outbreak and compare the microbiome to a timepoint after the outbreak.

## 2. Materials and Methods

### 2.1. Animal Population

All animal use was approved by the USMARC Animal Care and Use Committee. Data were collected in 2016 from cattle in advanced generations of the USMARC GPE (Germplasm Evaluation Program [21]) herd, Clay Center, Nebraska. The GPE cattle used within this study consisted of a variable fraction of 18 breeds: Angus, Hereford, Red Angus, Brahman, Charolais, Gelbvieh, Limousin, Simmental, Brangus, Beefmaster, Shorthorn, Maine Anjou, Santa Gertrudis, Chiangus, Salers, Braunvieh, South Devon, and Tarentaise. Animals included in the study were raised under similar management conditions, receiving standardized vaccinations and diets as described by Workman et al. [22]. Animals included in the study were divided into breeding groups in separate pastures at USMARC in Clay Center, NE. As the calves were part of the GPE populations, they were diverse in their breed composition, and breeding groups were primarily based on cow age, with breeds represented across each breeding group. Calves were sorted into groups of approximately 120 animals based on birth order and then sorted into breeding groups during initial vaccination (averaging approximately 45 days of age). After initial vaccination, the breeding groups had no direct contact with one another. 

### 2.2. BRDC Outbreak

At approximately 94 days (range: 74–126 days) of age, calves in one breeding group (n = 93 calves) were mass-treated for BRDC following the observation by the attending veterinarian that approximately 15–20% of the calves were displaying clinical signs of BRDC, including nasal discharge, respiratory distress, lethargy, and elevated temperature. At approximately 113 days (range: 81–141 days) of age, a second breeding group (n = 142 calves) were mass-treated for BRDC once 25–30% of the calves in that pasture were displaying clinical signs of BRDC. The mass treatment protocol for the calves was previously described by Workman et al. [22]. These groups that were mass-treated at the time of the outbreak will be identified as MT1 and MT2, respectively, throughout the manuscript. Based on the clinical signs and number of calves diagnosed as BRDC-positive, the MT2 group experienced a more severe outbreak compared to the MT1 group. 

### 2.3. Selection of Calves

For the data presented herein, 26 calves were selected from the MT1 group, and 30 calves were selected from the MT2 group for further analysis with rectal temperatures ranging from 37.9 to 41.8 °C. Selection was based on the extremes in rectal temperature. To provide a representation of the entire outbreak group, half of the selected calves had a higher temperature and half of the calves had a lower (normal) temperature. For the moderate outbreak group (MT1), 13 of the calves selected had a temperature over 39.4 °C (range: 39.4–40.3 °C) and 13 calves had a temperature less than 38.9 °C. For the more severe outbreak group (MT2), 15 of the calves selected had a temperature over 40 °C (range: 40–41.8 °C) and 15 calves had a temperature less than 38.9 °C. Two calves from the MT2 group were not present for sampling at the post-treatment timepoint and were therefore not included in the study. Calves with rectal temperatures that fell outside the temperatures designated as high and low (normal) were not included in the study.

### 2.4. Nasal Swab Collection

Nasal swab samples were collected from the upper nasal cavity of calves using 15.24 cm (6-inch) nasal swabs. Samples were collected as described by McDaneld et al. [23]. Briefly, the nasal swab was gently inserted into the nasal cavity at an approximate depth of 15 cm. The nasal swab was then rotated and removed. After collection of the sample, swabs were placed in a collection tube with transport buffer (buffered peptone water with 12% glycerol), drop-frozen in liquid nitrogen directly after collection, and stored at −80 °C. Nasal swabs were collected at the time of the outbreak when treatment occurred, and approximately 4 weeks post-treatment (average 130 days of age).

### 2.5. DNA Extraction and Library Preparation

Total DNA was extracted from each swab using a commercial kit (PowerSoil DNA Isolation Kit; Qiagen, Germantown, MD, USA) following the manufacturer’s instructions. Briefly, a single swab was removed from the original storage tube and placed in a 1.5 mL microcentrifuge tube with 750 μL phosphate-buffered saline along with 250 μL of the transport buffer from the original storage tube. The tube was vortexed at medium speed to aid in the removal of bacterial content from the nasal swab. The swab was then removed from the microcentrifuge tube, and the microcentrifuge tube was centrifuged at 15,000× *g* for 10 min to pellet the bacteria. The resulting pellet was then resuspended in 800 μL CD1 buffer (supplied with the kit) and processed using the kit following the manufacturer’s instructions. Extracted DNA samples were then quantified using a DNA spectrophotometer (DeNovix DS-11 FX Series; Wilmington, DE, USA). PCR-grade water was used as the negative control and processed with the other samples in the DNA extraction process to evaluate contamination in the kit reagents.

Amplicon library preparation of the samples was performed by PCR amplification using primers with index sequences as previously described that amplify hypervariable region 1 through 3 of the 16S rRNA gene, Appendix A [24]. The quality and quantity of the resulting 16S rRNA gene amplification were checked on a Fragment Analyzer (Advanced Analytical, Ankeny, IA, USA). By using indexed primers to amplify the 16S rRNA gene, individual samples were pooled into four sequencing runs and were sequenced using the 2 × 300 paired-end reads, the v3 600-cycle kit, and the Illumina MiSeq sequencing platform (Illumina, San Diego, CA, USA). 

### 2.6. Sequence Processing

Microbiome bioinformatics were performed using the QIIME2 pipeline (v. 2024.2) [25]. For all read files, primers were trimmed and any non-trimmed reads were filtered using cutadapt [26]. Trimmed sequence data were merged, quality-filtered, and denoised using the q2-dadad2 plugin [27]. All amplicon sequence variants (ASVs) were aligned with mafft (via q2-alignment) and used to construct a phylogeny with fasttree2 via q2-phylogeny [28,29]. Taxonomy was assigned to ASVs using the q2-feature-classifier classify-sklearn naïve Bayes taxonomy classifier against the Genome Taxonomy Database (GTDB) 202 release [30,31]. Data were then evaluated for common contaminants that may have originated from contaminated reagents or consumables during DNA extraction [32]. If bacterial genera of common contaminants were identified in the dataset, the second swab collected from the animal was extracted for DNA and subsequent 16S rRNA gene amplification. A prevalence filter was applied to remove ASVs with less than 10 counts across all samples or those present in less than 3 samples.

### 2.7. qPCR for Mycoplasma Species

The presence of *Mycoplasma* species *M. bovis*, *M. dispar*, and *M. bovirhinis* was evaluated by qPCR for the selected calves sampled at the time of the BRDC outbreaks and the same calves sampled later at preconditioning. The qPCR reactions were performed following the manufacturer’s instructions for SYBR Green (BioRad, Hercules, CA, USA). In each qPCR reaction, 2 μL of DNA from previously described DNA extractions was used, with 0.25 μL of the forward and reverse primer (10 mM), 12.5 μL SYBR Green master mix, and 10 μL molecular-grade H_2_O. Primers and cycling conditions for *M. bovis* were F: 5′-CTT GGA TCA GTG GCT TCA TTA GC-3′, R: 5′-GTC ACT ATG CGG AAT TCT TGG GT-3′, 95 °C for 3 min, 35 cycles of 95 °C for 15 s, 55 °C for 30 s, and 72 °C for 1 min. Primers and cycling conditions for *M. dispar* were F: 5′-TTA AAG CTC CAC CAA AAA-3′, R: 5′-GTA TCT AAA GCG GAC TAA-3′, 95 °C for 3 min, 35 cycles of 95 °C for 15 s, 54 °C for 30 s, and 72 °C for 30 s. Primers and cycling conditions for *M. bovirhinis* were F: 5′-GCT GAT AGA GAG GTC TAT CG-3′, R 5′-ATT ACT CGG GCA GTC TCC-3′, 95 °C for 3 min, 35 cycles of 95 °C for 15 s, and 60 °C for 1 min. Cycle threshold (Ct) values less than 35 were considered positive. The average Ct value and percentage of samples positive for each *Mycoplasma* species were determined for *M. bovis*, *M. dispar*, and *M. bovirhinis*.

### 2.8. Statistical Analysis

Downstream analyses were performed with QIIME2 using the q2-diversity plugin and R using functions from phyloseq v1.46.0, microbiome v1.23.1, vegan 2.6-4, lmerTest v3.1-3, and ANCOMBC v2.4.0 [33,34,35,36,37]. All analyses were conducted to estimate effects of outbreak group (MT1 and MT2), body temperature within group, and timepoint (outbreak, post-treatment) on community structure, and statistical significance was considered at *p* < 0.05. Alpha diversity metrics (using Shannon’s index and observed ASV) were calculated for all samples to assess the richness and evenness of microbial communities within each sample. A linear mixed model (lmer) was used to generate alpha diversity estimates for outbreak groups and timepoints while including individual animal to account for repeated measures. Beta diversity analysis using Bray–Curtis distances and Unweighted Unifrac distances were completed to compare the microbial community composition between samples and identify differences in species presence, abundance, and phylogenetic relationships across groups. This was achieved using a PERMANOVA test with the adonis2() function from the vegan package with 999 permutations and including animalID, outbreak group, and timepoint as terms. To visualize these differences, data were ordinated using a principal coordinate analysis (PCoA) graph for both Bray–Curtis dissimilarities and Unweighted Unifrac distances. To identify variation in taxa abundance, taxa counts were transformed to relative abundance and grouped at the phylum and genus levels. Taxa that could not be identified at that level were categorized as unknown. Differential abundance for individual taxa at the genus level was analyzed for both outbreak group and timepoint using Analysis of Compositions of Microbiomes with Bias Correction (ANCOM-BC) to identify which specific microbes significantly changed in abundance across conditions, revealing taxa potentially associated with disease or temporal shifts in the microbiome. ANCOM-BC estimates the unknown sampling fraction and corrects for bias introduced by sampling fraction using a log linear regression model to identify taxa that are differentially abundant according to the variable of interest and controls for the false discovery rate. qPCR data for specific *Mycoplasma* species were analyzed using ANOVA with a fixed effect of sampling time by outbreak group (subgroup class) to analyze preplanned contrasts. Because several qPCR samples showed no signal for specific strains, a Ct value of 40 was assigned to test differences among subclasses.

## 3. Results and Discussion

An evaluation of the upper nasal microbiome was completed in pre-weaned calves at the time of a BRDC outbreak. Samples were collected at the time of outbreak for two groups of calves and a later timepoint (preconditioning) to identify the bacterial populations present in the upper nasal cavity and changes post-treatment. Variation among bacterial profiles was identified for both the outbreak group and later timepoint. No differences were observed in microbial abundance or diversity related to the body temperature of the calf at time of outbreak, and therefore this variable will not be discussed further. 

### 3.1. Alpha Diversity

At the time of the outbreaks, the group of calves that experienced the more severe outbreak (MT2) had the lowest microbial community diversity across both groups and timepoints (Figure 1). This lower alpha diversity, as measured by Shannon’s diversity index, indicates that MT2 calves had fewer bacterial taxa overall at the time of the disease outbreak. Although both MT1 and MT2 had a significant increase in alpha diversity between the outbreak and post-treatment timepoints (*p* < 0.001), microbial diversity remained lower for MT2 compared to MT1 at post-treatment (*p* < 0.001), suggesting that while treatment allowed some recovery of the microbial community, the more severely affected calves had not yet regained the microbial richness seen in MT1. Lower diversity in the nasal microbiome of calves has previously been associated with BRDC, reinforcing the idea that microbial community structure may play a role in susceptibility to disease [15,38]. 

Additionally, pathogenic organisms associated with BRDC have been identified as common inhabitants of the nasal microbiome in pre-weaned calves and feedlot steers [12,14]. It is likely that BRDC reduces the diversity of the nasal microbiome and allows pathogenic bacteria to become more prevalent, creating a microbial environment that favors disease progression. The greater nasal microbial diversity of the moderately severe-BRDC animals compared to the highly severe-BRDC animals suggest that community diversity may not only influence the likelihood of animals contracting BRDC but also contribute to their ability to recover from BRDC. This is supported by the alpha diversity metrics in Figure 1, demonstrating the separation between MT1 and MT1 at both outbreak and post-treatment timepoints, highlighting the importance of microbial diversity as a potential biomarker for disease severity and resilience. 

Variation in alpha diversity among calves increased from the outbreak to post-treatment for MT2 but decreased for MT1 (Figure 1; Appendix A). This indicates that for MT2, there were low-diversity and high-diversity calves that maintained their level of diversity between the time of the outbreak and 4 weeks post-treatment. Conversely, MT1 calves more consistently increased their alpha diversity, with decreased variation in alpha diversity among calves post-treatment. In other words, there was a consistent increase in diversity within calves post-treatment. 

### 3.2. Beta Diversity

Interestingly, between-calf differences in microbial community composition (beta diversity) resulted in calves clustering differently between MT1 and MT2, with more overlap between sampling times within MT1 or MT2 than between these two groups (Figure 2). This indicated that calves within the same outbreak group shared more similar microbial community structures over time compared to the calves from the other outbreak group. There were more outlier calves for MT2 than for MT1 primarily at 4 weeks post-treatment, highlighting the heterogeneity in recovery of the microbial community among calves experiencing a more severe outbreak. These findings emphasize the important interactions between microbial community composition during and after mass treatment. Regarding the interaction, the microbial community for MT1 upon mass treatment was similar to MT2 4 weeks post-treatment whereas MT2 upon mass treatment and MT1 4 weeks post-treatment each had distinct communities (Figure 3A), suggesting that microbial community dynamics and composition differ depending on outbreak severity.

In terms of evaluating microbial community structure or distance between sampling timepoints and outbreak groups (beta diversity), all groups were significantly different across timepoints (*p* < 0.001) (Figure 2; Appendix A). Both Bray–Curtis and unweighted UniFrac analysis revealed that while MT1 and MT2 both experienced a BRDC outbreak, the microbial community composition varied substantially between the two groups. Considered alongside alpha diversity metrics, this supports the theory of a complex relationship between microbial diversity and the occurrence and severity of BRDC. At the time of the outbreak, the group that experienced the more severe outbreak (MT2) not only had lower microbial diversity but varied in community composition when compared to the more moderate BRDC outbreak group (MT1). As expected, community structure differed between the time of the outbreak when the animals were mass-treated and later at four weeks post-treatment. However, the two outbreak groups continued to maintain distinct microbial community structures after treatment. These differences indicate that both the severity of the outbreak and individual calf responses influence the structure of the nasal microbiome over time. The differences in microbial community structure were further explored using ANCOMBC to identify the distinct taxa associated with each group and time. 

### 3.3. Abundance

Overall, significant differences in microbial profiles were identified between both the outbreak groups and sampling timepoints (Figure 3). The phyla that were identified as making up more than 1% of the total abundance were *Bacillota* (synonym *Firmicutes*; 56.2%), *Bacteroidota* (synonym *Bacteroidetes*; 22.3%), *Pseudomonadota* (synonym *Proteobacteria*; 7.0%), *Actinomycetota* (4.4%), *Chloroflexota* (2.2%), and *Acidobacteriota* (1.6%). *Bacillota* is often noted as one of the primary phyla associated with the nasal microbiome in cattle [13,20,34]. However, in the current study, *Bacillota* accounted for more than half of the nasal microbiota present, which is much greater than the 10–30% of the total abundance that has previously been reported [12,13,34]. Similarly, *Pseudomonadota* is often cited as one of the most prominent phyla of the bovine nasal microbiota but made up less than 10% of the microbes detected in the current study. This deviation from previously reported abundances highlights the dynamic nature of the nasal microbiome and the importance of monitoring variation in relation to disease outbreaks. 

For the outbreak sampling timepoints, *Mycoplasmopsis* was the predominant genus present. The genera *Mesomycoplasma* and *Caviibacter* were also present in greater abundance at the time of the outbreak compared to 4 weeks post-treatment. *Mycoplasmopsis* and *Mesomycoplasma* were formally referred to collectively as the genus *Mycoplasma* but have been reassigned as separate genera [39]. Species within the genera *Mycoplasma* previously associated with BRDC include *M. bovis*, *M. dispar*, and *M. bovirhinis* [14,20,40]. Under current taxonomic nomenclature, *M. bovis* and *M. bovirhinis* belong to the genus *Mycoplasmopsis* while *M. dispar* is a member of the genus *Mesomycoplasma* [41]. The genus *Mycoplasmopsis* was present in high abundance across all sampling points, particularly the time of the outbreak, and was especially abundant in the group of calves that experienced the more severe outbreak (MT2). Although *Mycoplasmopsis* sp. decreased in MT2 at preconditioning, *Mesomycoplasma* sp. increased in abundance post-treatment. *Mycoplasma* sp. are often found in high abundance in the nasal microbiome regardless of health status [12,38]. The results from this study support previous suggestions that stressors or changes in an animal environment can cause dysbiosis within the nasal microbiome, allowing opportunistic *Mycoplasma* sp. to thrive. These findings emphasize the importance of understanding temporal shifts in the nasal microbiota, as interventions that support or restore homeostasis in microbial communities may improve disease outcomes. 

In addition to *Mycoplasma* sp., *Histophilus* sp. was identified during the two outbreak timepoints at 1–5% of relative abundance. This is the first occurrence of our laboratory identifying *Histophilus* sp. at an abundance greater than 0.1% at any timepoint prior to or after weaning [23]. The occurrence of *Histophilus* sp. at higher abundance during outbreaks indicates its potential role as a co-pathogen in BRDC and supports the need to monitor both primary and secondary microbial populations during disease events. These data suggest that *Mycoplasma* sp. and *Histophilus* sp. may be the predominant bacterial genera associated with the two BRDC outbreaks. Collectively, these results highlight the relevance of characterizing both the composition and dynamics of the nasal microbiome for understanding disease risk and guiding management strategies in pre-weaned calves.

Regarding differential abundance at the genus level, we identified significant changes (*p* < 0.01) in several key pathogenic organisms and enrichment for a larger number of genera previously associated with core organisms of the nasal respiratory system in calves (Figure 4) [13,38]. These results combined with previous studies emphasize how BRDC outbreaks can disrupt the normal nasal microbial community, allowing certain pathogenic or opportunistic organisms to dominate and commensal taxa to fluctuate [12,13,15,40]. The number of species enriched both between outbreak groups and post-treatment aligns with the noted increase in alpha diversity metrics for those samples, with MT1 having greater enrichment over MT2 and post-treatment having more enriched organisms compared to the outbreak timepoint. This correspondence between alpha diversity and genus-level enrichment underscores the significance of microbial community recovery following disease or intervention, as more diverse communities may better resist colonization by pathogenic organisms. 

All organisms that were depleted post-treatment have been well documented for their association with BRDC [12,14,15,40]. Although not all BRDC-affiliated bacterial genera were characterized as having high overall abundance, the significant decrease in the pathogenic BRDC-associated organism between the time of the outbreak and post-treatment indicated that they contributed to disease occurrence. *Mannheimia haemolytica* is often identified as a prominent organism associated with BRDC outbreaks [15,40]. Only *Mycoplasmopsis* sp. were enriched in MT2 compared to MT1. Because the MT2 group experienced a more severe outbreak of BRDC compared to MT1, this implies that *Mycoplasmopsis* sp. abundance may have been a primary factor in the development and severity of BRDC for these calves. Because MT2 experienced a more severe BRDC outbreak than MT1, the enrichment of *Mycoplasmopsis* sp. may indicate its role as a primary driver of disease severity, either directly or indirectly through pathogenic mechanisms or indirectly by interacting with other organisms. These findings reinforce the importance of monitoring both abundance and shifts in key BRDC-associated taxa in pre-weaned calves. These dynamics may aid in predicting outbreak severity and guide targeted intervention strategies, reducing morbidly, mortality, and antibiotic use. 

### 3.4. qPCR for Mycoplasma Species

With *Mycoplasma* sp. being the most abundant genus in calves evaluated at the time of the BRDC outbreaks, specific *Mycoplasma* species associated with the BRDC outbreaks were identified. Specific *Mycoplasma* species evaluated included *M. bovis*, *M. dispar*, and *M. bovirhinis*, which were quantified by qPCR. *Mycoplasma bovis* is traditionally associated with BRDC [42,43]. However, other *Mycoplasma* species including *M. dispar* and *M. bovirhinis* have been identified in the microbiome of cattle with BRDC [12,20]. For the data presented herein, *M. bovirhinis* was the predominant species present (Ct value < 23.0) with *M. bovis* and *M. dispar* having a lower abundance with a Ct value > 29.0 at both BRDC outbreak timepoints (Figure 5; Appendix A), suggesting that *M. bovirhinis* is the predominant *Mycoplasma* species associated with the BRDC outbreaks.

For *M. dispar*, the abundance and percent of calves that tested positive decreased between timepoints for MT1 but remained constant for MT2. Overall, MT1 had a greater reduction in both abundance and percent for positive calves compared to MT2 between the time of outbreak and 4 weeks post-treatment. Although there are some differences between the current results and those previously reported by Workman et al. [22], which involved this group of animals, this difference is likely due to the scope of that work. The primary focus of the previous work was evaluating the effect of Bovine corona virus (BCV) on BRDC incidence in pre-weaned beef calves. Although qPCR was used to detect *M. haemolytica*, *P. multocida*, *H. somni*, and *M. bovis*, they did not include *M. dispar* or *M. bovirhinis*. Additionally, because the main objective of that study was to evaluate BCV, transport media and qPCR assays differed between studies.

## 4. Conclusions

The results of this study indicate that reduced microbial diversity and the overrepresentation of pathogenic *Mycoplasma* sp., particularly *M. bovirhinis*, are closely linked to disease occurrence in pre-weaned calves. Previous research characterizing microbial communities associated with BRDC has largely focused on the feedlot phase. Less is known regarding microbial community dynamics prior to the feedlot phase, and to our knowledge, this is the first study to evaluate a large group of calves during multiple BRDC outbreaks and at a timepoint after the outbreak. Overall, a decrease in diversity and a dominance of *Mycoplasma* sp. characterized the nasal microbial communities at the timepoint of the BRDC outbreaks. While both groups recovered a more diverse upper nasal microbiome post-treatment, there remained differences in diversity between the groups. This indicates that microbial community species abundance and diversity have a role in the severity of BRDC in pre-weaned calves. These findings highlight microbial diversity as both a potential indicator of disease risk and target for interventions. Importantly, improving strategies to stabilize or restore the nasal microbiome may not only reduce BRDC-related losses in cattle but also reduce the need for antimicrobial use, thereby lowering the risk of antimicrobial resistance with implications for both animal and human health. We recognize that we have a relatively small sample size for the mass-treated groups. Additional factors to consider when evaluating the results include the variable fractions of breed in each calf and the possibility that more subclinical calves not displaying symptoms may be present in the younger mass-treated group (MT1). Overall, further evaluation of changes in the upper nasal microbiome prior to weaning will improve our understanding of the relationship between the microbiome and the incidence of BRDC.

## Figures and Tables

**Figure 1 animals-15-02914-f001:**
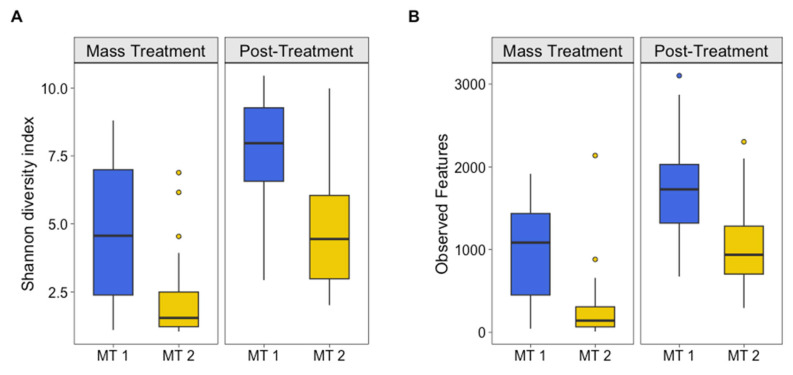
Alpha diversity. (**A**) Shannon’s diversity index and (**B**) observed features for groups (MT1, mass-treated BRDC outbreak group 1 (blue column); MT2, mass-treated BRDC outbreak group 2 (yellow column)) at the time of a BRDC outbreak and approximately 4 weeks post-treatment.

**Figure 2 animals-15-02914-f002:**
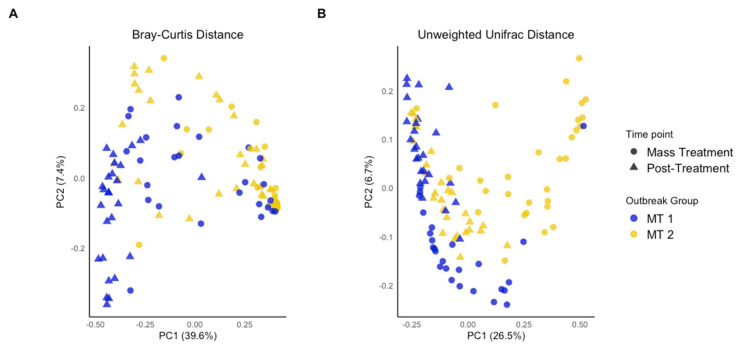
PCoA plot of beta diversity analysis. Principal component analysis of bacterial communities among groups (MT1, mass-treated BRDC outbreak group 1 (blue dots); MT2, mass-treated BRDC outbreak group 2 (yellow dots)) and timepoints (outbreak (dots) and approximately 4 weeks post-treatment (triangles)) calculated using the Bray–Curtis method (**A**) and unweighted Unifrac distances (**B**).

**Figure 3 animals-15-02914-f003:**
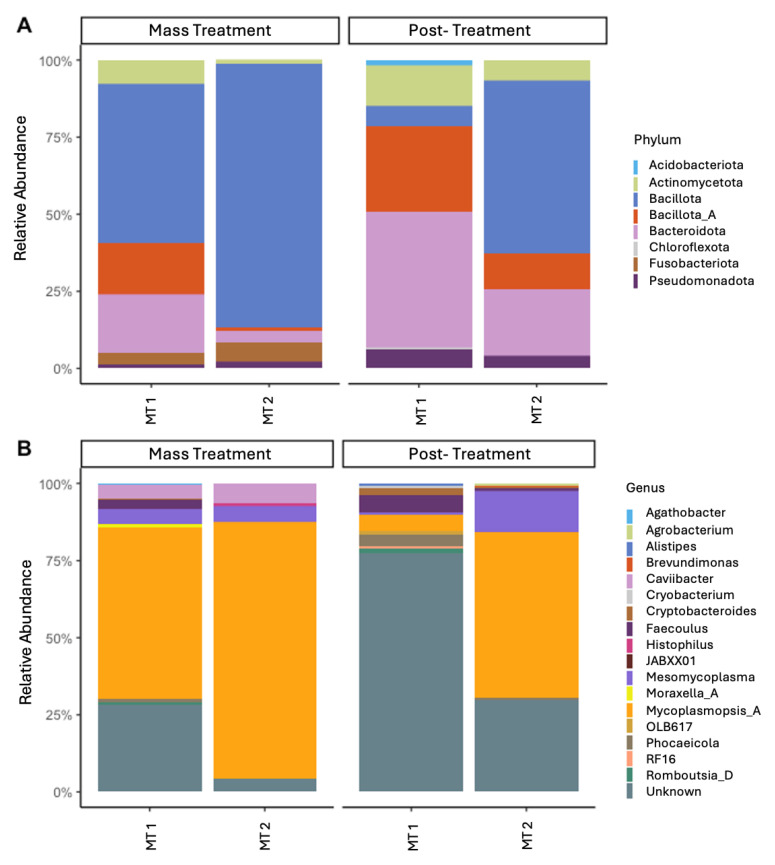
Relative abundance of bacterial present at a relative abundance of 5% or greater at the level of phylum (**A**) and genus (**B**). 16S rRNA gene profiles of bacterial genera were evaluated in calves during two separate BRDC outbreaks. Samples were collected from two groups (MT1, mass-treated BRDC outbreak group 1; MT2, mass-treated BRDC outbreak group 2) at the time of BRDC outbreak and approximately 4 weeks post-treatment.

**Figure 4 animals-15-02914-f004:**
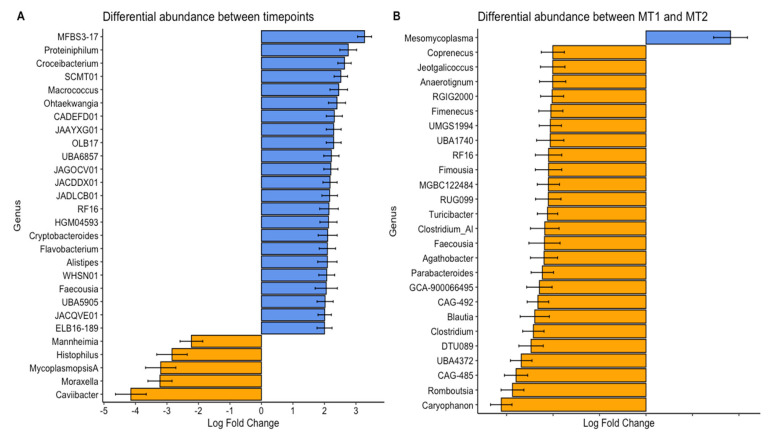
Differential abundance of bacterial genera between two BRDC outbreak groups (**A**) and timepoints (**B**). Samples were collected from two groups (MT1, mass-treated BRDC outbreak group 1; MT2, mass-treated BRDC outbreak group 2) at the time of BRDC outbreak and approximately 4 weeks post-treatment.

**Figure 5 animals-15-02914-f005:**
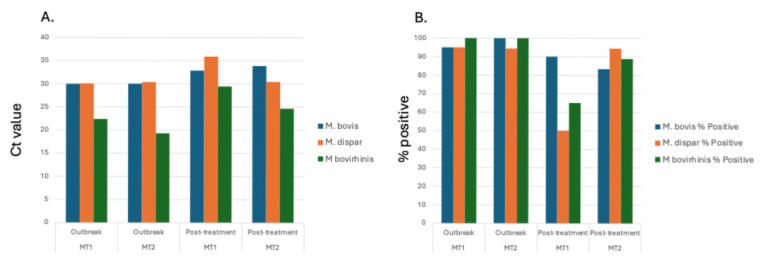
Average abundance and positivity percentage of respiratory pathogens in calves. Average Ct value (**A**) and percentage of samples that tested positive (**B**) for *Mycoplasma* species *M. bovis*, *M. dispar*, and *M. bovirhinis* for calves.

## Data Availability

The original contributions presented in the study are publicly available. This data can be found at https://github.com/ana2277/BRDC.PreweanedCalves (created 10 December 2024); Nasopharyngeal sequences were deposited in the NCBI sequence read archive (SRA) database under Bioproject: PRJNA1144906, BioSamples: SRX25828196–SRX25828087.

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
