# Peer review of "Evaluation of Nasal Microbial Communities of Beef Calves During Pre-Weaning Outbreak of Bovine Respiratory Disease"

_animals, 2025, doi:10.3390/ani15192914_

Round 1
Reviewer 1 Report
Comments and Suggestions for Authors
Dear authors, I believe that the article provides important information in the context of bovine health, but in the current version there are several areas for improvement. Therefore, I offer several observations that, once addressed, will improve the quality of the article for its subsequent reconsideration for publication in the journal:
- As reported by iThenticate, the similarity percentage is 36%, which is very high. I suggest lowering the percentage to improve the quality of the article and avoid plagiarism.
- The "Abstract" contains important information, but I suggest restructuring it to include subsections such as background, methodology, results, and conclusions. This will allow readers to properly understand what was done in the research.
- The section "1. Background" should be renamed to "1. Introduction", I consider that it is brief and more information should be added so that readers have all the necessary information to understand the research topic, I suggest that everything related to BRDC be mentioned in more detail, such as etiology, symptoms, diagnosis and treatment, in addition to mentioning normal microbiota bacteria and pathogenic bacteria associated with BRDC, the role of BRDC in animal health and human health, highlighting its importance in the cattle industry.
- In section "2. Materials and methods" I recommend adding subsections, for example: "2.1 Animal Population", "2.2 BRDC Outbreak", etc. According to subsection "Selection of calves" should be 2.3 and 2.4 should be "Nasal Swab Collection". In section "2.3 Selection of calves" I recommend that the selection criteria for the calves from which nasal samples were taken be explained in more detail, in addition to mentioning exclusion criteria, this will allow readers to properly understand the criteria used for the selection of calves.
- Section "2.5 DNA Extraction and Library Preparation" should briefly explain the instructions for DNA extraction using the kit used. Additionally, it is mentioned that the primers with index sequences for performing the PCR were previously described, but I cannot find that information in the article. This will allow readers to properly understand the methodologies used in the research.
- In section "2.6 Sequence Processing" I suggest explaining in more detail the process of evaluating data to detect common contaminants from reagents during DNA extraction, this will allow readers to properly understand the bioinformatics analyses performed in the research.
- In section "2.7 qPCR for Mycoplasma Species" I suggest that the characteristics of the reaction, quantity of reagents, etc., be explained in more detail, this will allow readers to properly understand what is related to this methodology.
- Section "2.8 Statistical analysis" contains a lot of information that corresponds to previous sections. I suggest mentioning the type of analysis performed, the justification for the use of statistical tests, the level of significance, and the statistical software used in the research.
- Personally, I prefer when the results are presented together with the discussion, but I haven't seen the journal allow it, so I suggest creating separate sections, "3. Results" and "4. Discussion," to comply with the journal's requirements.
- I recommend that subsections be added to section "3. Results" for better classification of the results obtained in the research, according to Figure 1. I consider that the results related to the Shannon Diversity Index can be explained more, because its explanation in the text is brief and its importance in the research is not very clear. In addition, I consider that Table 1 does not provide relevant information in the context of the research, I believe that Table 1 does not provide much relevant information in the context of the research, so I suggest giving greater importance to the other results, this will improve the quality of the article.
- I believe the explanation of the results in the PCoA graph (Figure 2) could be better. The impact of these results on the research is not entirely clear, so I believe it should be explained a little more in order to emphasize its importance in the article. Additionally, like Table 1, I believe that Table 2 does not provide much relevant information in the context of the research and I suggest giving greater importance to other results.
- Figure 3 looks blurry, I suggest replacing it with a higher quality one, this will allow readers to better analyze the results.
- Based on the results of the qPCR for Mycoplasma species, I consider that a table is not the best option for presenting the results. I recommend creating one or more graphs that would allow readers to interpret the results more appropriately.
- As I mentioned earlier, I recommend that section "4. Discussion" be elaborated; furthermore, the discussion carried out in this version is not entirely adequate, so I recommend that the authors discuss much better the results obtained in this research with results obtained by other researchers, this will allow the reader to properly understand the impact of their results in the context of research in other countries around the world.
- As mentioned above, the "Conclusions" section should be number 5 and could be improved. I recommend briefly including the results of the research and its potential impact on animal and human health, if it adequately meets the research objectives, to further emphasize its results and its impact on public health in the United States and around the world.
Author Response
We appreciate the reviewers comments. All comments have been addressed below with the comment numbered and the response of the authors following. Changes in the revised manuscript are in red text.
Reviewer 1:
- As reported by iThenticate, the similarity percentage is 36%, which is very high. I suggest lowering the percentage to improve the quality of the article and avoid plagiarism.
Response: Upon evaluation of the iThenticate report, the highest similarity (11%) was calculated for sections of the introduction and methods. This similarity was to openprarie.sdstate.edu, which is the senior authors (Dr. Amy Abrams) PhD dissertation. It is common for PhD dissertations in the Animal Science department at South Dakota State University to be structured that each chapter is a manuscript ready for submission to a journal for the introduction and methods sections. As a result, the chapter of Dr. Amy Abrams’ dissertation and this current manuscript are highly similar. Additionally, the second highest similarity of 9% is to one of our previous publications in Frontiers. This similarity is for the methods section for the PCR amplification for the 16S rRNA libraries and the qPCR. These are our standard methods for these procedures.
- The "Abstract" contains important information, but I suggest restructuring it to include subsections such as background, methodology, results, and conclusions. This will allow readers to properly understand what was done in the research.
Response: Based on the layout style guide of Animals (2.4 Abstracts), authors should follow the style guide of a structured abstract without using headings. To follow this guideline, we have not restructured the abstract to include subsection headings.
- The section "1. Background" should be renamed to "1. Introduction", I consider that it is brief and more information should be added so that readers have all the necessary information to understand the research topic, I suggest that everything related to BRDC be mentioned in more detail, such as etiology, symptoms, diagnosis and treatment, in addition to mentioning normal microbiota bacteria and pathogenic bacteria associated with BRDC, the role of BRDC in animal health and human health, highlighting its importance in the cattle industry.
Response: Section 1. Background has been changed to Introduction. Additional information has also been added to the background section. Lines 66, 72-75, and 80-87 have been added. While there currently is no literature that associates bovine respiratory disease with human health, it has been proposed that alternatives to antibiotic treatment for respiratory disease is of interest. This has been included in the Introduction.
- In section "2. Materials and methods" I recommend adding subsections, for example: "2.1 Animal Population", "2.2 BRDC Outbreak", etc. According to subsection "Selection of calves" should be 2.3 and 2.4 should be "Nasal Swab Collection". In section "2.3 Selection of calves" I recommend that the selection criteria for the calves from which nasal samples were taken be explained in more detail, in addition to mentioning exclusion criteria, this will allow readers to properly understand the criteria used for the selection of calves.
Response: As suggested, subsections have been added for clarity. To provide more detail for the criteria of calf selection, Lines 132 and 139-140 have been added.
- Section "2.5 DNA Extraction and Library Preparation" should briefly explain the instructions for DNA extraction using the kit used. Additionally, it is mentioned that the primers with index sequences for performing the PCR were previously described, but I cannot find that information in the article. This will allow readers to properly understand the methodologies used in the research.
Response: Lines 146 and 152-159 have been added to provide detail for DNA extraction. To clarify the PCR amplification section, Supplemental Table 1 has been added to list the indexed primers that were used in the PCR amplification for sequence libraries.
- In section "2.6 Sequence Processing" I suggest explaining in more detail the process of evaluating data to detect common contaminants from reagents during DNA extraction, this will allow readers to properly understand the bioinformatics analyses performed in the research.
Response: Lines 160-162 were added to provide further clarification for detection of possible contaminants.
- In section "2.7 qPCR for Mycoplasma Species" I suggest that the characteristics of the reaction, quantity of reagents, etc., be explained in more detail, this will allow readers to properly understand what is related to this methodology.
Response: Appreciate the reviewer pointing out this was missing. Lines 188-189 have been added for more detail of qPCR reactions.
- Section "2.8 Statistical analysis" contains a lot of information that corresponds to previous sections. I suggest mentioning the type of analysis performed, the justification for the use of statistical tests, the level of significance, and the statistical software used in the research.
Response: As suggested, text has been added for justification of the statistical tests used. Level of significance and statistical software used has also been added. Lines 205, 206-207, 210-212, and 221-222 were added.
- Personally, I prefer when the results are presented together with the discussion, but I haven't seen the journal allow it, so I suggest creating separate sections, "3. Results" and "4. Discussion," to comply with the journal's requirements.
Response: We agree that we prefer combination of the results and discussion. Based on the layout style guide of Animals (3.1 Overall Structure), authors may choose to have Results and Discussions as one or two sections. We have chosen to combine the results and discussion into one section.
- I recommend that subsections be added to section "3. Results" for better classification of the results obtained in the research, according to Figure 1. I consider that the results related to the Shannon Diversity Index can be explained more, because its explanation in the text is brief and its importance in the research is not very clear. In addition, I consider that Table 1 does not provide relevant information in the context of the research, I believe that Table 1 does not provide much relevant information in the context of the research, so I suggest giving greater importance to the other results, this will improve the quality of the article.
Response: As suggested, subsections have been added. Additionally, text has been added for further explanation of the Shannon Diversity Index. Lines 241-242 have been added. Table 1 has been moved to supplemental tables as we believe these data support the corresponding figures and are of importance to the manuscript.
- I believe the explanation of the results in the PCoA graph (Figure 2) could be better. The impact of these results on the research is not entirely clear, so I believe it should be explained a little more in order to emphasize its importance in the article. Additionally, like Table 1, I believe that Table 2 does not provide much relevant information in the context of the research and I suggest giving greater importance to other results.
Response: As suggested, text has been added for further explanation of the PCoA graph. Lines 277-279 have been added. Table 2 has been moved to supplemental tables as we believe these data support the corresponding figures and are of importance to the manuscript.
- Figure 3 looks blurry, I suggest replacing it with a higher quality one, this will allow readers to better analyze the results.
Response: Figure 3 has been replaced with a higher resolution figure.
- Based on the results of the qPCR for Mycoplasmaspecies, I consider that a table is not the best option for presenting the results. I recommend creating one or more graphs that would allow readers to interpret the results more appropriately.
Response: Table 3 has been converted to Figure 5. The X and Y axis can be switched if this would be easier to interpret. Table 3 has been converted to a supplemental table (Supplemental Table 4) as the table format may be preferred by others.
- As I mentioned earlier, I recommend that section "4. Discussion" be elaborated; furthermore, the discussion carried out in this version is not entirely adequate, so I recommend that the authors discuss much better the results obtained in this research with results obtained by other researchers, this will allow the reader to properly understand the impact of their results in the context of research in other countries around the world.
Response: As suggested, the discussion has been expanded to better summarize and elaborate on the results. Lines 245-247, 248-249, 256-259, 280-282, 286-287, 300-301, 322-323, 338-340, 344-346, 348-350, 363-365, 368-371, and 380-386 have been added.
- As mentioned above, the "Conclusions" section should be number 5 and could be improved. I recommend briefly including the results of the research and its potential impact on animal and human health, if it adequately meets the research objectives, to further emphasize its results and its impact on public health in the United States and around the world.
Response: The conclusion has been numbered, and additional text has been added. Lines 422-424 have been added.
Reviewer 2 Report
Comments and Suggestions for Authors
An approximate difference of 5% (20% in MT1 vs 25% in the MT2) is not that much in a practical application. Could have been more subclinical in the lighter group too. Need to discuss this limitation well in the discussion. Also wouldn't the groups be confounded by age? Collecting samples from another group which did not break with BRDC at the similar time would have been valuable as a potential control group. Overall, very interesting article.
Line 100: Please clarify as state only 9 calves had BRDC, but then in nasal swab collection describe more samples. If this was post-weaning, would be recommended to move later in the materials and methods. Also would recommend creating on section to describe age, weight, how animals were diagnosed as sick in the feedlot phase.
Author Response
Appreciate the comments of the reviewer. Response to the reviewer are listed by number with the response following. All the changes in the manuscript are in red text.
1. An approximate difference of 5% (20% in MT1 vs 25% in the MT2) is not that much in a practical application. Could have been more subclinical in the lighter group too. Need to discuss this limitation well in the discussion. Also wouldn't the groups be confounded by age? Collecting samples from another group which did not break with BRDC at the similar time would have been valuable as a potential control group. Overall, very interesting article.
Response: Lines 439-443 have been added to state the limitations of the study to include sample size and subclinicals in the MT1 group. For the age of the animals in each group, the difference between the average ages is less than 20 days (113 days – 94 days). While it is possible that the first group may have more subclinical cases that were not displaying symptoms, we do not have a way to determine this. A sentence has been added to the manuscript to propose number of subclinicals may be different between the two groups.
We are in agreement with the reviewer that a control group of calves that did not break with BRDC would have benefited the study. However, after treating the group of calves for BRDC, it was not possible to sample control calves as that was not part of the approved experimental outline. Additionally, our concern was that running a group of healthy calves through for sampling after the mass treatment group of calves would have increased stress for the healthy group of calves.
2. Line 100: Please clarify as state only 9 calves had BRDC, but then in nasal swab collection describe more samples. If this was post-weaning, would be recommended to move later in the materials and methods. Also would recommend creating on section to describe age, weight, how animals were diagnosed as sick in the feedlot phase.
Response: The authors have collectively decided to remove the sentence about the 9 animals that were diagnosed with BRDC in the feedlot. We originally added this sentence to provide follow up for the calves in the study after they were weaned at the feedlot. However, as no data is provided in the manuscript for these calves at the feedlot, we have deleted this sentence to avoid confusion. Please advise if it would be preferred to leave this information in the manuscript.
Reviewer 3 Report
Comments and Suggestions for Authors
Dear Authors,
the study addresses an important issue, namely BRDC in calf in the context of the nasal microbiome. However, the work has certain shortcomings. It is not explained whether the animals came from a single farm or from multiple farms. If this was a typical feedlot where animals from different farms are gathered, this would also affect the respiratory tract microflora. Potentially, different breeds may have varying susceptibility to BRDC. The cause of BRDC occurrence has not been established, nor whether it was related to the differing course of disease observed in the study groups. The results are interesting, but the group of animals from which samples were collected is relatively small (26 and 30), and this is further complicated by the presence of several cattle breeds within the study group.
In summary, the study is valuable but requires more detail in the areas indicated above.
Author Response
Appreciate the reviewers comments. Comments are numbered and the author response follows. All changes to the manuscript are in red text.
1. The study addresses an important issue, namely BRDC in calf in the context of the nasal microbiome. However, the work has certain shortcomings. It is not explained whether the animals came from a single farm or from multiple farms. If this was a typical feedlot where animals from different farms are gathered, this would also affect the respiratory tract microflora. Potentially, different breeds may have varying susceptibility to BRDC. The cause of BRDC occurrence has not been established, nor whether it was related to the differing course of disease observed in the study groups. The results are interesting, but the group of animals from which samples were collected is relatively small (26 and 30), and this is further complicated by the presence of several cattle breeds within the study group.
Response: As stated in Lines 102-104, all the cattle used in this study are from our population of the USMARC GPE (Germplasm Evaluation Program; Retallick et al., 2017). As a result, all the calves came from a single source. Additional information has been added to Line 110 to clarify this. The GPE population is population that consists of a variable fraction of the top 18 breeds that are registered in the United States. These include Angus, Hereford, Red Angus, Brahman, Charolais, Gelbvieh, Limousin, Simmental, Brangus, Beefmaster, Shorthorn, Maine Anjou, Santa Gertrudis, Chiangus, Salers, Braunvieh, South Devon, and Tarentaise. As a result, we did not include breed in the analysis. Given the number of breeds, we do not have the power to fit breed in the model. Additional information in Lines 104 and 110-111 has been added to further clarify that all the calves were from the GPE population at USMARC and of multiple breed composition.
Round 2
Reviewer 1 Report
Comments and Suggestions for Authors
Dear authors, I consider that all the observations shared previously have been satisfactorily addressed, the only observation is the following:
- Themes and subthemes do not have points, so the point that the subthemes have in section "2. Materials and methods" and "3. Results and Discussion" must be removed.
I believe the quality of the article has improved and I approve it for publication in the journal. Congratulations.
Author Response
Thank you for your time and effort in reviewing the manuscript. As a result it is more cohesive. We will consult the editor for proper formatting of the themes and sub themes to have them in the correct format.
Reviewer 2 Report
Comments and Suggestions for Authors
Authors addressed my concerns. Nice manuscript.
Author Response
Thank you for your comments, it resulted in a more cohesive manuscript.